# Redefining Food Sources: Exploring the Disconnect between Goat Farming and Its Perceived Sustainability—A Case Study in Chitima, Mozambique

**Martín del Valle M** [1,2], **Roberto Hernández** [3], **Lisa Boden** [2] **and José Luis Riveros** [1,*]

¹ Department of Animal Sciences, Faculty of Agronomy and Forest Engineering, Pontificia Universidad Católica de Chile, Santiago 7820436, Chile; martin.delvalle@ed.ac.uk

² Global Academy of Agriculture and Food Systems, The University of Edinburgh, Midlothian EH25 9RG, UK; lisa.boden@ed.ac.uk

³ Department of Environmental Sciences and Renewable Natural Resources, Faculty of Agricultural Sciences, Universidad de Chile, Santiago 8820808, Chile; callevarela@gmail.com

* Correspondence: jlriverosf@uc.cl; Tel.: +56-2-23544142

**Abstract:** The objective of this study was to analyze the disconnect between goat farming and its perception as a food source and determine if this is related to the way farmers value goats. We conducted a questionnaire of 1 open and 15 closed questions among (*n* = 23) goat producers in Bairro Boroma, Province of Tete, Mozambique, and six in-depth interviews with stakeholders of the local goat meat trading chain. The results show that goats have an economic value more than a nutritional value, meaning they preferred trading them instead of allocating them for their own consumption. Conformation and size characteristics were more relevant when buying/selling an animal. All goat producers sold live animals in their household's backyard, at an average price of USD 19.3 ± 4.6/per animal with an average weight of 20.23 ± 3.41 kg. Finally, goat producers preferred to increase their number of goats rather than cattle due to the ease of husbandry, amount of meat obtained from the carcass and reduced risk of meat spoilage post-slaughter because of their smaller size. The latter is essential within an environment lacking access to drinkable water and electric energy to support cold chain management in meat storage.

**Keywords:** cultural value; goats; food security; famine; Mozambique

## 1. Introduction

Animal production in low- and middle-income countries (LMICs) is mainly related to self-sufficiency, and it is considered a priority to meet the existing needs of individuals, families, and local pastoral communities. Together with the increasing demand for livestock products due to population growth, rapid urbanization and increasing incomes [1], it offers significant nutritional and economic alternatives for households located in areas where the availability, physical access, and affordability of other sources of food is scarce ([2] p. 13).

However, this activity does not necessarily lead to better health outcomes, as protein and micronutrient deficiencies and low scores on dietary diversity are still issues for people who belong to these pastoral communities, increasing their chances of not achieving financial independence and leaving them vulnerable to ongoing cycles of poverty [3].

Since these communities are highly knowledgeable and experienced at managing livestock within these contexts, it is important to explore other social and cultural sustainability determinants that could explain why such a disconnect between animal production and nutritional deficiencies exists [4].

### 1.1. Food Sources and Sustainability

When relating food and sustainability, although most studies are focused on the "health–environment" duality [5], recent approaches have started considering other context-

specific concerns as sustainability drivers, such as the affordability and socio-cultural acceptability of diets at the global, regional, local, and individual level [6,7]. From this sustainability perspective, it becomes critical to understand the way local food sources, either animal or vegetable, are perceived in order to better understand how they can tackle regional challenges such as food insecurity.

### 1.2. Goats as a Resource against Poverty

Goats and humans have lived together in a mutualist relationship for more than 10,000 years [8,9]. Even it is a species distributed transversally in different latitudes, according to FAO STAT [10], they are mostly concentrated in Africa (43.4%), Asia (51.4%) and the Americas (3.5%), where they play a unique role in supporting vulnerable communities and can represent a useful tool for overcoming poverty with special value as a "mobile bank" [11,12]. These kinds of contexts are mostly dominated by resilient native biotypes, which have developed morphological and physiological adaptations that make them valuable to farmers even in harsh conditions such as excessive temperatures and a lack of feed and water, among others [13–16]. Low production costs and fast reproduction, in addition to rapidly increasing the size of the herd, allow herders to value or treat livestock not just as sources of good quality milk, meat, and fiber, but also as currency and security for loans to acquire other goods or adapt in emergency situations, contributing to household food and health security [17,18].

In this paper, we assess the perceived sustainability of goat farming in tackling food insecurity by analyzing goat production, sales, the use of cash income from goat sales and the perceived value of goats as a food source in Mozambique. We also discuss how farmers perceive "hunger" and adapt or mitigate the risks of food scarcity and lack of dietary diversity. We conducted this study using mixed quantitative and qualitative approaches that allowed us to understand how local farmers value their goats a whole and in comparison to their livestock in a context characterized by its social vulnerability, especially in terms of food insecurity.

## 2. Materials and Methods

### 2.1. The Case Study Context—Chitima, Mozambique

The Republic of Mozambique is a country in southeast Africa, with a population of close to 32 million people [19]. Approximately 46.61% of Mozambicans lived below poverty line in 2014; however, it is estimated that because of the COVID-19 pandemic, that number increased to 63.3% in 2021 [20]. According to the World Food Programme [21], 80% people in Mozambique cannot afford an adequate diet, and 42% of children are stunted because of food insecurity.

The present study was conducted in *Bairro Boroma* (Boroma neighborhood) located at Chitima (15°43′58.78″ S; 32°46′7.27″ E), governmental headquarter of Cahora Bassa District, Province of Tete. The rural city of Chitima had in 2010 an estimated population of 20,135 inhabitants [22]. On the other hand, the Province of Tete, located in the northwest of the country, has a total area of 100,742 km$^2$, a tropical–dry climate with an average annual rainfall of 600 mm and a total population of 2,137,700 inhabitants [22].

### 2.2. Surveys and Interviews

A preliminary meeting was held in January 2017 with the local informants, who acted as mediators between the researchers and the participants, of *Bairro Boroma* to discuss the objectives, methodology, inclusion criteria and benefits that this research might have for participants. In this meeting, the questionnaire was reviewed by the local authorities to validate the tool and to determine the best way to ask questions to ensure the inhabitants of *Bairro Boroma* understood them. After that, and using the methodology discussed with the leaders, a workshop with the community was performed to share with them what was agreed with their representatives and clarify any remaining uncertainties. This preliminary engagement was aimed at involving the community in the research process.

After the study received ethical approval by the Pontificia Universidad Católica de Chile, farmers were recruited by the leaders of the *Bairro Boroma's* community and informed consent was sought before participation. Written questionnaire surveys, developed in Portuguese and performed in *Nyungwe*, consisted of 1 open and 15 closed questions on goat production and the valuation of goats to better understand their motivations for goat production over other activities and the cultural and/or economic determinants that could increase goats' value. Producer distribution information was collected by geo-referencing location with GPS. Information was collected from those in each household who oversaw goat production, the average age and primary language spoken (i.e., Portuguese-speaking). Survey questions are shown in Table 1.

**Table 1.** Survey questions applied to *Bairro Boroma* goat farmers.

| | Head of Household Information |
|---|---|
| 1 | Gender |
| 2 | Age |
| | **Valorization of goats** |
| 3 | What is a goat to you? |
| 4 | Which of the following characteristics is the most important when choosing a goat? |
| | Sex |
| | Size |
| | Color |
| | Number of births |
| | Conformation |
| | Horns |
| | Others |
| 5 | Did your father/grandfather valued the same? |
| | Yes/No |
| | If not, why? |
| 6 | Does the value of the animal change when there is more grass? |
| | Yes |
| | No |
| 7 | Does the value of the animal change according to the number of pregnancies? |
| | Yes |
| | No |
| 8 | Does the value of the animal change according to its conformation? |
| | Yes |
| | No |
| 9 | Does the value of the animal change according to its color? |
| | Yes |
| | No |
| 10 | Would you like to have more cattle or goats? Why? |
| | Cattle |
| | Goat |
| 11 | Do you prefer beef or goat meat? |
| | Beef |
| | Goat |
| 12 | Do other animals have the same importance? |
| | Yes, which? |
| | No, why? |

**Table 1.** *Cont.*

| Head of Household Information | |
| --- | --- |
| Goat production and selling | |
| 13 | Number of |
| | Adult males |
| | Adult females |
| | Male goatlings |
| | Female goatlings |
| 14 | Selling price |
| | Un-castrated male |
| | Castrated male |
| | Female that has given birth |
| | Female that has not given birth |
| 15 | Do you sell the entire animal or by pieces? |
| 16 | Where do you sell? |
| | Backyard |
| | Market |
| | Slaughterhouse (*Talho*) |

Besides households, marketing locations for goat meat, such as the slaughterhouse and market, were also geo-referenced, and their managers were interviewed in order to know both where they were located and the local prices of meat in Chitima's meat value chain. Specifically, they were asked for the value of an un-castrated male, a castrated male, a female that has given birth, and a female that has not given birth.

Survey information was stored in spreadsheets and then analyzed using descriptive statistics to obtain frequencies, means and standard deviations. Additionally, six in-depth interviews were carried out with key community actors to understand how they perceive "hunger times" and the importance of goat-keeping in dealing with it. Interviews were conducted in Portuguese and in one case it was necessary to translate questions to *Nyungwe* with the help of a local translator. As with the surveys, each interview was performed following active informed consent.

Data were tabulated and analyzed through Microsoft Excel© and presented as frequencies in terms of percentages, averages, and standard deviations. In order to safeguard the data from accidental or deliberate access when it was out of the University premises, we stored it on memory sticks with encrypted password protection.

## 3. Results and Discussion

### 3.1. Socio-Demographic Information

Of the total number of households (*n* = 25) dedicated to goat production in *Bairro Boroma*, 23 answered the survey (92% response rate). The two households that did not participate were not present at home or in the neighborhood at the times of the study visit. Questionnaires took an average of 18 ± 5 min to be completed. There were several challenges related to data collection, including long distances between our headquarters to some of the study households of *Bairro Boroma*, due to the brush during that rainy season being rugged and difficult to travel on, and the presence of snakes on the road to get there.

Most households (70%, *n* = 16 households) were managed by adult men, who are commonly in charge of all activities related with livestock and meat commercialization. Women represented a smaller percentage of goat producers (26% *n* = 6), and children were responsible for 4% (*n* = 1) of households. According to [23], who described a similar situation in Tanzania, this reality can be due to a strong cultural background in favor of men over women, especially in livestock husbandry. The average age of producers, both men and women, was 43 ± 15.7 years. According to the World Health Organization [24], the life expectancy in Mozambique is 56 for men and 59 for women.

Figure 1 shows the distribution of goat producers' settlements, which, except for household 19 (S19), are ordered following the course of a river that separates them from

Chitima village, where the slaughterhouse (*talho*) and market (*mercado*) are located. It is important to note that all producers share pasture areas where, as recognized by them, there is a constant risk of loss and mixture of the herd.

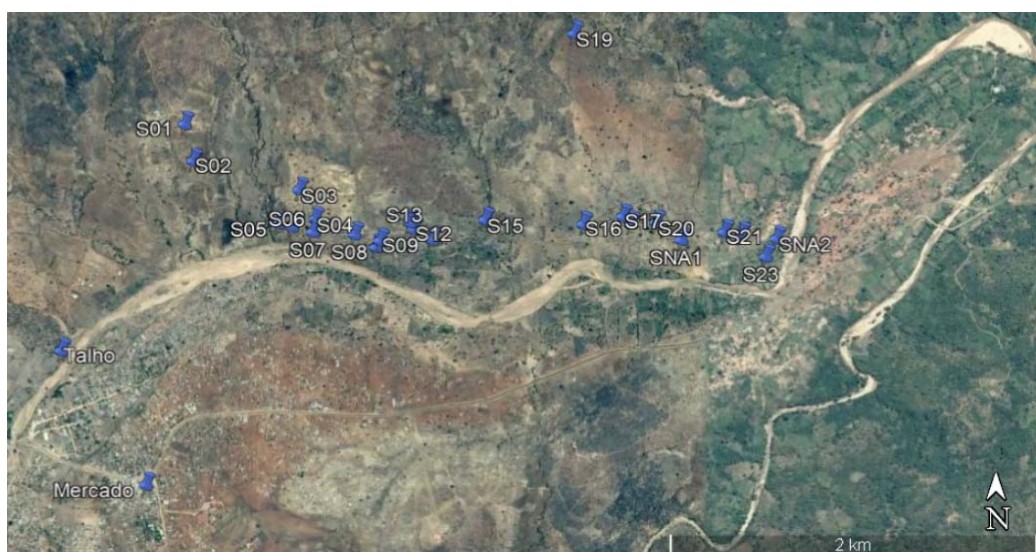

**Figure 1.** Distribution of goat producers in *Bairro Boroma*, Village of Chitima, District of Cahora Bassa, Province of Tete, Mozambique.

### 3.2. How Goats Are Avalued in Times of Hunger

"Times of hunger" (*tempos de fome*) were described by *Bairro Boroma* actors as a period between October to March in which, besides humans, animals, pastures, and forest suffer especially from water shortage. This systemic view is related to the fact that food production is considered by them as a process in which "men manage nature for its own benefit", as mentioned by one respondent. However, hunger is not only explained by water shortage. To a 68-year-old male actor, it is also important to consider how knowledge to cope with these times is used. According to this respondent, in terms of water availability, "times of hunger" were less frequent before than nowadays, especially because effective precipitation used to be more frequent. However, because of the independence and civil war period, many people were constantly moving from one place to another, making it impossible to establish in a certain place and practice agriculture. A 59-year-old female respondent remembered how children from her district presented serious ulcers related to undernourishment induced by constant migrations.

*Bairro Boroma* goat producers believe that goats are a "source of richness" that, in the case of necessity, can be sold for food or used as collateral to obtain loans. In this sense, it was indicated by producers that the perceived income from sales goes mainly to purchasing other food group items such as corn flour, beans, and some green leaf vegetables. Most producers (43.5%, *n* = 10) consider goats foremost as animals for meat and milk production. However, near equal numbers of participants (39.1%, *n* = 9) were unable to explain the main purpose for keeping goats. A small number (17.6%, *n* = 4) believe that goats are the "herd's mother". Almost all producers (95.7%, *n* = 22), indicated that the way they perceive goats is the same as their parents or grandparents did. This last point is particularly relevant for a case study in perceived sustainability, since it shows almost no changes in the way different generations understand it within these contexts. However, it also highlights the possibility that there are diverse ways of perceiving time and that what academia/policy understands as "sustainability" can be understood differently in other contexts by using different words and/or having distinct behaviors.

Most *Bairro Boroma* goat producers (47.8%, *n* = 11) think goats are a better livestock option than other farm animals because of their facility to work, their salability, their shorter reproductive cycle, and their rapid multiplication. Additionally, the relative ease of the

slaughtering process and the amount of meat obtained offers advantages over cattle. Goats are also preferred over cattle (39.1%, *n* = 9) because of the difficulties associated with beef processing (i.e., the greater size and difficulties associated with meat preservation due to the lack of electric energy). The latter may lead to greater obligations to share meat with neighbors without guarantee of reciprocity. Conversely, cattle are also considered important livestock because of their utility in soil preparation for crops, transport, and as assets that can be sold to purchase other goods for households. Something that was not mentioned by any producer in this matter was the relevance of water access, which is a critical aspect considering the intake differences between goats and livestock, and is widely documented as an advantage that goat production has over other major species [13–16]. Further research would be valuable to better understand water governance in such contexts, where there is a constant tension for its access between humans, animals, and the environment. A total of 13.1% of households showed no preference between these two species. All producers mentioned that hens were valuable due to egg production, either for household consumption or selling. Regarding preferences related to meat, 52.2% (*n* = 12) of producers declared to prefer goat meat mainly due to its taste over beef. Those who preferred beef (26.1%, *n* = 6) indicated that a "not so strong smell" was the main reason to choose it over goat meat. Finally, the remaining 21.7% (*n* = 5) indicated no preference.

When choosing a goat, either at the market or at any household backyard, most producers (82.4%, *n* = 19) suggested that size (big or small animals) and conformation (based on their body condition) of goats are the most important attributes (Figure 2). When selling an animal in backyards, though, all producers sell the entire goat instead of by pieces. Breed (only local breeds), health (if the goat looked strong or weak), early kidding (referred to the age goats give birth for the first time), and number of births (number of kids born per birth) were considered by 17.6% (*n* = 4) of total respondents to be less important. As complementary information, and due to the time constraint of the research teams spent living in Chitima, we know that goats never received supplementary feed and that occasionally some producers received veterinary assistance as part of *Cahora Bassa* district agricultural aid and farming support. Thus, in average, an animal would cost USD 19, while an un-castrated male would cost USD 17, a castrated male USD 24, a female who has given birth USD 22, and a female who has not given birth USD 14.

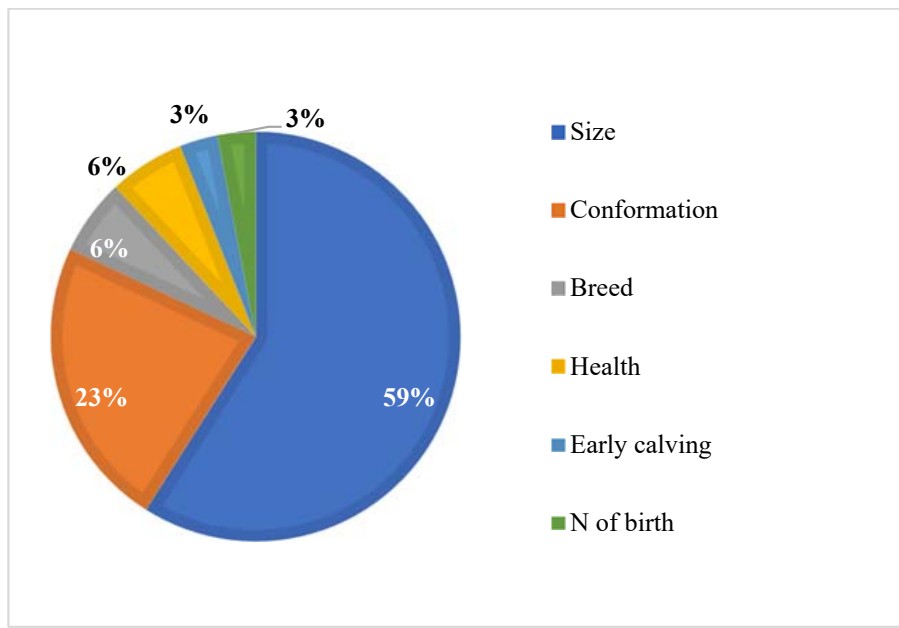

**Figure 2.** Goat attribute importance when choosing an animal to goat producers of *Bairro Boroma*.

The producers surveyed, indicated that castrated males have a value close to 140% higher compared to the un-castrated males because of the tendency to become fat and increase in live weight, an important factor at the time of sale (Figure 2). Other determinants affecting goat commercial value are presented in Figure 3. Pasture availability associated with goat price may be explained by the possibility of having bigger animals, supporting the data in Figure 2. In relation to the number of parities, the additional value for having at least one birth is explained by the security of not having problems when getting pregnant. Confirming what is indicated in Figure 2, a good conformation is directly related to the animal's final price, for both female and male animals.

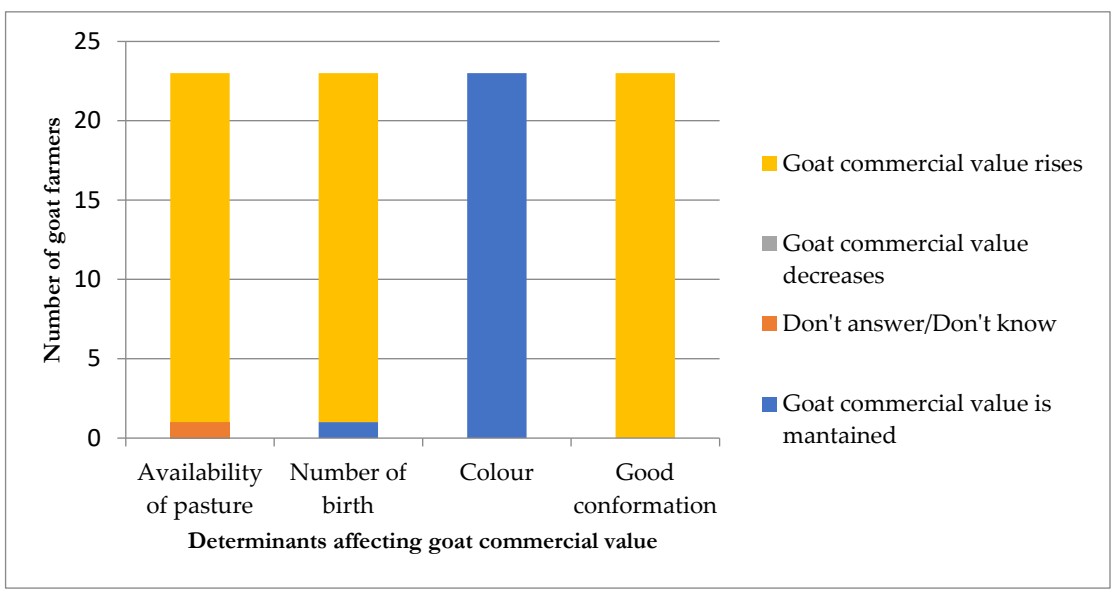

**Figure 3.** Determinants affecting goat commercial value, *Bairro Boroma*, Chitima, Mozambique.

Goat value chain involves the interaction between goat producers, and meat sellers from the *mercado* or the *talho*, shown in Figure 1, where it is possible to buy meat as cuts, not as in households' backyards, where the whole animal is sold. Besides, in these places, sanitary conditions are maintained in contrast to what happens at the backyard level. As described before, the average price among different type of animals is USD 19/animal, while at the market each kg of goat meat is sold at USD 1.43. Goat meat sellers told us that from each goat it is possible to obtain 22 kg of meat, which would mean USD 31.7 per goat, a 33.2% uplift on what producers obtain at their backyards. Although these figures may suggest that there is an important financial benefit that it is not being taken advantage of, other aspects such as distance to the market, logistics, slaughtering knowledge, and market fees, among others, should also be considered when looking at sustainable uses and the perception of edible products. Also related to the economic value, this important income difference can play a relevant role in the context of food insecurity, since most of the money that comes from goat sales is used to buy other food products that complement the household nutritional needs.

Considering this study's results, a next step could be to invest in social/technological innovation that allows the use of local knowledge to preserve goat (and other species) meat and address food insecurity challenges. This could benefit the community from the point of view of the nutritional value of the animal and the impact on the health of the households, particularly in this type of context. An example of this type of food and local knowledge-related innovation is a meat-drying technique known as *chinkui*, whose case study was developed by the authors of this study and is documented in [4]. Additionally, this case report could give some insights into how to continue to study how farmers value their productive resources, especially from a gender perspective considering the world-wide importance women have on household food security. We propose to escalate

the questionnaire to the Cahora Bassa district to include more households and be able to account for gender differences.

## 4. Conclusions

To Bairro Boroma goat producers, goats are perceived and valued as a source of income more than a food source, and they are part of a commercial enterprise of Chitima, which includes selling at the household level and buying meat in external places such as the local market and the slaughterhouse.

To live through "times of hunger" seems to be natural for inhabitants of *Bairro Boroma*, so it becomes fundamental to explore alternatives that allow the community to prevent, adapt or mitigate the risks of food scarcity and lack of dietary diversity. Relatedly, goat production in northern regions appears as a sustainable and culturally appropriate resource to deal with the above.

Goat valuation itself considers "visual" aspects such as size and conformation rather than attributes considered as technical such as the breed or number of births. Additionally, this type of production is more valued than cattle to Bairro Boroma goat producers despite not being useful for draught power, and this seems to be mainly explained by the amount of meat obtained and the ease of handling of smaller volumes of meat post-slaughter. This last aspect indicates a possibility of development in terms of improving the conditions to preserve meat.

Finally, we believe that the results from this small farming community could relate to a broader picture of similar contexts by highlighting the importance of cultural, technical, and commercial determinants of food security and livestock supply chains in a context of rural social vulnerability.

**Author Contributions:** M.d.V.M., R.H. and J.L.R.: conceptualization and methodology; M.d.V.M. and J.L.R.: investigation and writing; R.H. and J.L.R.: supervision, review, and editing; L.B.: review and editing. All authors have read and agreed to the published version of the manuscript.

**Funding:** This research was funded by the "Fondo Chile" fund (AGCID-PNUD—2015) and the Master program of Sistemas de Producción Animal of the Pontificia Universidad Católica de Chile.

**Institutional Review Board Statement:** The study was conducted in accordance with the Declaration of Helsinki, and approved by the Institutional Review Board of Pontificia Universidad Católica de Chile (161212007/16-12-2016).

**Informed Consent Statement:** Informed consent was obtained from all subjects involved in the study.

**Data Availability Statement:** Not applicable.

**Acknowledgments:** We thank the Centro O Viveiro NGO in Chitima, Mozambique, for receiving us as family and allowing us to share with them during the last two years. Also, to Alan Duncan from The University of Edinburgh, for his review and comments on this work.

**Conflicts of Interest:** The authors declare no conflict of interest.

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
