# Peer review of "Redefining Food Sources: Exploring the Disconnect between Goat Farming and Its Perceived Sustainability—A Case Study in Chitima, Mozambique"

_sustainability, doi:10.3390/su151411071_

Round 1

Reviewer 1 Report

The manuscript by Valle et al. presents a case study conducted in a village in Mozambique, focusing on assessing the farmers' perceptions of goat production. The study findings highlight an important aspect, revealing that farmer value goats primarily as a source of income rather than for household consumption. This finding emphasizes the potential sustainability of goat production, as a farming practice that provides consistent income can contribute to the long-term viability and stability of the agricultural system. It underscores the significance of understanding farmers' perspectives and priorities in shaping sustainable farming practices. The manuscript was well-written and presented in a manner that was easy to follow. Since the manuscript is a case report, the small sample size should not be a major concern. However, there are some feedback and suggestions provided below to further improve the manuscript:

Title: Please use the phrase “case study” instead of “ study case”

Line 41, 51: Please consider adding a reference to support the statement.

Consider moving the objectives of the study to the last paragraph of the introduction section. This will help provide a clear transition between the background information and the specific aims of the research.

The description of the country in the last paragraph appears out of place and interrupts the flow of the introduction. It would be more appropriate to incorporate relevant information about the country within the broader context of the study, such as the socio-economic factors or farming practices.  

Line 216: Line 216: It would be more appropriate to use the term "early kidding" instead of "calving".

Line 242: The goat sellers told the researchers having meat weighing approximately 22 kg from each goat, while the reported pre-slaughter weight was also around 22 kg (line 223). However, when considering the concept of dressing percentage, these values do not align. This discrepancy should be addressed and clarified in order to ensure the accuracy and coherence of the results.

The manuscript would benefit from providing a more thorough discussion and interpretation of the findings. While the results are presented clearly, it is important to delve deeper into the implications and significance of the findings.

The in-text references in the manuscript do not adhere to the style guidelines of the journal, particularly regarding the format of citations. Instead of using the author's name and year in parentheses, it is recommended to use numerical references enclosed in brackets. For example, "(Delgado, 2005)" should be presented as "[2]". This revision should be applied consistently throughout the manuscript.

Overall, the manuscript is promising, and with the suggested improvements, it has the potential to make a valuable contribution to the existing literature.

Author Response

Thank you for the unvaluable suggestions you have made on our manuscript. Kindly, find below our comments to each one of them, and attached the reviewed document.

  • Title: Please use the phrase “case study” instead of “ study case”.
    • R: Now amended.

  • Line 41, 51: Please consider adding a reference to support the statement.
    • R: Statement removed due to changes in the introduction

  • Consider moving the objectives of the study to the last paragraph of the introduction section. This will help provide a clear transition between the background information and the specific aims of the research.
    • R: Now amended.

  • The description of the country in the last paragraph appears out of place and interrupts the flow of the introduction. It would be more appropriate to incorporate relevant information about the country within the broader context of the study, such as the socio-economic factors or farming practices.
    • R: R1 and R2 suggestions regarding this matter were included.

  • Line 216: Line 216: It would be more appropriate to use the term "early kidding" instead of "calving".
    • Now amended.

  • Line 242: The goat sellers told the researchers having meat weighing approximately 22 kg from each goat, while the reported pre-slaughter weight was also around 22 kg (line 223). However, when considering the concept of dressing percentage, these values do not align. This discrepancy should be addressed and clarified in order to ensure the accuracy and coherence of the results.
    • R: We removed the information from the previous study, as was not accurate since it was explaining a context with different and controlled conditions.

  • The manuscript would benefit from providing a more thorough discussion and interpretation of the findings. While the results are presented clearly, it is important to delve deeper into the implications and significance of the findings.
    • R: more discussion in L178-173; L193-200; L240-244.

  • The in-text references in the manuscript do not adhere to the style guidelines of the journal, particularly regarding the format of citations. Instead of using the author's name and year in parentheses, it is recommended to use numerical references enclosed in brackets. For example, "(Delgado, 2005)" should be presented as "[2]". This revision should be applied consistently throughout the manuscript.
    • R: Now amended.

Reviewer 2 Report

I think the idea is good but you need to do more to produce a manuscript that can be published in a scientific journal.

Introduction: Difficult to follow. I suggest you leave out developed countries stuff and focus on developing countries so that you can zero in on your case study.  

Suggestion Line 28 to 51

”Poor rural communities in Africa, Asia, Latin America, and Oceania practice traditional pastoral production. Mozambique is one of the countries with ...........................These communities are highly knowledgeable and experienced at managing livestock in  environments with limited feed and water resources. For these communities livestock production offers nutritional benefits in areas where availability, physical access, and affordability of other sources of food is scarce (Chappell, 2018) [1, p. 13] and economic benefits due to increasing demand for livestock products due to population growth, rapid urbanization and increasing incomes (Delgado, 2005) [2]. However, it seems the nutritional benefits of livestock production are not being realized in many of these communities. Protein and micronutrient deficiencies and low scores on dietary diversity condition growth are still common in these communities (Neumann et al., 2002) [3]. It is important to understand how nutritional outcomes are affected by production, low consumption of meat and milk due to livestock producers prefering to sell the animals, and use of cash to buy other type of goods and services.”

Line 57

”In this sense, is...” could be ”From this sustainability perspective....”

Line 77 to 78

”In this paper, we analyze the disconnect between goat farming and its perception as a food source and determine if this is related to the way farmers value goats.”

An objective should be very clear so that readers can easily understand the paper. This is not very clear.

I suggest you write something like this:

”In this paper, we assess the perceived sustainability of goat farming in tackling food insecurity by analysing goat production, sales, use of cash income from goat sales and perceived value of goats as a food source in Mozambique.”

I would recommend you to remove section 1.3 and write a few sentences about Mozambique in the introduction and a few sentences at the start of the method section.  

Materials and methods

Table 1: The questionnaire can be in the supplementary, there is no need to have it in the main paper.

-You could use regression as a method with nutritional outcomes as the dependent variable and indicator of perceived sustainability

-Define constructs e.g. perceived value of goats as a food source adopting theories e.g. consumer perceived value theory (Sheth et al., 1991; Zeithaml, 1988)

-Would expect a scale (likert) on variables such as perception of goats as a food source

-Would expect to see calculations of reliability and validity e.g. cronbach alpha etc

-Would expect to see use exploratory analysis i.e. the explore dimensions that make up constructs and confirmatory analysis

-Would expect a large sample size including households with different nutritional outcomes e.g. high and low scores on dietary diversity

Author Response

Thank you for the unvaluable suggestions you have made on our manuscript. Kindly, find below our comments to each one of them, and attached the reviewed document.

  • Introduction: Difficult to follow. I suggest you leave out developed countries stuff and focus on developing countries so that you can zero in on your case study.
    • R: Now amended.

  • Suggestion Line 28 to 51

”Poor rural communities in Africa, Asia, Latin America, and Oceania practice traditional pastoral production. Mozambique is one of the countries with ...........................These communities are highly knowledgeable and experienced at managing livestock in  environments with limited feed and water resources. For these communities livestock production offers nutritional benefits in areas where availability, physical access, and affordability of other sources of food is scarce (Chappell, 2018) [1, p. 13] and economic benefits due to increasing demand for livestock products due to population growth, rapid urbanization and increasing incomes (Delgado, 2005) [2]. However, it seems the nutritional benefits of livestock production are not being realized in many of these communities. Protein and micronutrient deficiencies and low scores on dietary diversity condition growth are still common in these communities (Neumann et al., 2002) [3]. It is important to understand how nutritional outcomes are affected by production, low consumption of meat and milk due to livestock producers prefering to sell the animals, and use of cash to buy other type of goods and services.”

R: We took some insights from this paragraph suggestion.

  • Line 57: ”In this sense, is...” could be ”From this sustainability perspective....”
    • R: Now amended.

  • Line 77 to 78: ”In this paper, we analyze the disconnect between goat farming and its perception as a food source and determine if this is related to the way farmers value goats.”
    • R: Suggestion below included.

  • An objective should be very clear so that readers can easily understand the paper. This is not very clear. I suggest you write something like this: ”In this paper, we assess the perceived sustainability of goat farming in tackling food insecurity by analysing goat production, sales, use of cash income from goat sales and perceived value of goats as a food source in Mozambique.”
    • R: Suggestion included.

  • I would recommend you to remove section 1.3 and write a few sentences about Mozambique in the introduction and a few sentences at the start of the method section.
    • R: Suggestion included.

  • Materials and methods

  • Table 1: The questionnaire can be in the supplementary, there is no need to have it in the main paper.
  • R: Ok, we will ask the editors to move it as supplementary material.

  • You could use regression as a method with nutritional outcomes as the dependent variable and indicator of perceived sustainability.
  • Define constructs e.g. perceived value of goats as a food source adopting theories e.g. consumer perceived value theory (Sheth et al., 1991; Zeithaml, 1988).
  • Would expect a scale (likert) on variables such as perception of goats as a food source.
  • Would expect to see calculations of reliability and validity e.g. cronbach alpha etc.
  • Would expect to see use exploratory analysis i.e. the explore dimensions that make up constructs and confirmatory analysis.

R: We value all these suggestions, but since this is a study limited to small number of participants, we believe the methodology proposed allows us to tackle the paper’s objectives.

  • Would expect a large sample size including households with different nutritional outcomes e.g. high and low scores on dietary diversity.

R: Since this is a case report, we think the sample size should not be a major concern.

Round 2

Reviewer 2 Report

-Author: "Thank you for the unvaluable suggestions you have made on our manuscript."

What do you mean with 'unvaluable'? 

Introduction

-Line 53-54: Please change to "....it is important to explore other social and cultural sustainability determinants that could explain why such disconnect between animal production and nutritional deficiencies exists [4]."

Results and discussion

-Figure 3: What is the unit of the y-axis?

-I am still missing details about what happens to the income from goat sales. Can you add a statement?

-What is the implication of having goats being valued for economic reasons? I mean for development organization who want to improve nutritional outcomes in Chitima? You should discuss such things in your discussion. Maybe it part of chinkui, its not very clear. 

Conclusion

I think the conclusion should start with this statement: 

"To Bairro Boroma goat producers, goats are perceived and valued as a source of income more than a food source, and they are part of a commercial enterprise of Chitima, which includes selling at the household level and buying the meat in external places such as the local market and the slaughterhouse."

Line 254-264: This (below) should be part of the discussion and not the conclusion.

"Related to the above, a next step could be to invest in social/technological innovation that allows the use of local knowledge to preserve goat (and other species) meat and address food insecurity challenges. This could benefit the community from the point of view of the nutritional value of the animal and the impact on the health of the households, particularly in this type of context. An example of this type of innovation is chinkui, whose case study was developed by the authors of this study and is documented in [4]. Also, this case report could give some insights on how to continue research on how farmers value their productive resources, especially from a gender perspective considering the world-wide importance women have on household food security. We propose to escalate the questionnaire to the Cahora Bassa district to count with more households and be able to account for gender differences."

Author Response

Thanks again for your comments and suggestions. Kindly find below our responses to each one of them.

  1. Author: "Thank you for the unvaluable suggestions you have made on our manuscript." What do you mean with 'unvaluable'? 

R: We meant your suggestions made us see the gaps our manuscript still had and the opportunities for improvement.

  1. Introduction

Line 53-54: Please change to "....it is important to explore other social and cultural sustainability determinants that could explain why such disconnect between animal production and nutritional deficiencies exists [4]."

R: Now amended.

  1. Results and discussion

-Figure 3: What is the unit of the y-axis?

N: Both axes now named.

-I am still missing details about what happens to the income from goat sales. Can you add a statement?

            R: Now amended. Detailed in L168-171.: . “In this sense, it was indicated by producers that the perceived income from sales was goes mainly to purchasing other food groups items such as corn flour, beans, and some green leaves vegetables”.

-What is the implication of having goats being valued for economic reasons? I mean for development organization who want to improve nutritional outcomes in Chitima? You should discuss such things in your discussion. Maybe it part of chinkui, its not very clear. 

R: Now amended. We included the following statement in L240-243: “Also related to the economic value, this important income difference can play a relevant role in a context of food insecurity, since most of the money that comes from goat sales is used to buy other food products that complement the household nutritional needs”.

Related to chinkuy, we modified the statement as follows in L248-250:  An example of this type of food and local knowledge-related innovation is a meat drying technique known as chinkui, whose case study was developed by the authors of this study and is documented in [4]

Conclusion

I think the conclusion should start with this statement: 

"To Bairro Boroma goat producers, goats are perceived and valued as a source of income more than a food source, and they are part of a commercial enterprise of Chitima, which includes selling at the household level and buying the meat in external places such as the local market and the slaughterhouse."

R: Now amended.

Line 254-264: This (below) should be part of the discussion and not the conclusion.

"Related to the above, a next step could be to invest in social/technological innovation that allows the use of local knowledge to preserve goat (and other species) meat and address food insecurity challenges. This could benefit the community from the point of view of the nutritional value of the animal and the impact on the health of the households, particularly in this type of context. An example of this type of innovation is chinkui, whose case study was developed by the authors of this study and is documented in [4]. Also, this case report could give some insights on how to continue research on how farmers value their productive resources, especially from a gender perspective considering the world-wide importance women have on household food security. We propose to escalate the questionnaire to the Cahora Bassa district to count with more households and be able to account for gender differences."

R: Now amended.
